# A Closed Cavity Ultrasonic Resonator Formed by Graphene/PMMA Membrane for Acoustic Application

**DOI:** 10.3390/mi14040810

**Published:** 2023-04-01

**Authors:** Jing Xu, Graham S. Wood, Enrico Mastropaolo, Peter Lomax, Michael Newton, Rebecca Cheung

**Affiliations:** 1The School of Engineering, Institute for Integrated Micro and Nano Systems, University of Edinburgh, Edinburgh EH9 3FF, UK; 2AAC Technologies (Scotland) Limited, Edinburgh EH3 8EG, UK; 3The Acoustics and Audio Group, University of Edinburgh, Edinburgh EH8 9DF, UK

**Keywords:** graphene, ultrasound, MEMS, resonator

## Abstract

A graphene/poly(methyl methacrylate) (PMMA) closed cavity resonator with a resonant frequency at around 160 kHz has been fabricated. A six-layer graphene structure with a 450 nm PMMA laminated layer has been dry-transferred onto the closed cavity with an air gap of 105 μm. The resonator has been actuated in an atmosphere and at room temperature by mechanical, electrostatic and electro-thermal methods. The (1,1) mode has been observed to dominate the resonance, which suggests that the graphene/PMMA membrane has been perfectly clamped and seals the closed cavity. The degree of linearity of the membrane’s displacement versus the actuation signal has been determined. The resonant frequency has been observed to be tuned to around 4% by applying an AC voltage through the membrane. The strain has been estimated to be around 0.08%. This research puts forward a graphene-based sensor design for acoustic sensing.

## 1. Introduction

Graphene has aroused much attention from the research and industrial community since it was discovered [1] due to its outstanding electrical and mechanical properties, namely its ultra-high Young’s modulus and mechanical strength [2], superior electron mobility [3] and super-low mass density. The application of graphene has provided a path to a new class of resonators and sensors in the past 15 years, such as pressure sensors [4,5,6,7], electromechanical actuators [8], resonators [9,10,11,12,13,14,15,16], microphones [17,18,19,20], nanodrums [21,22] and bio-sensors [23].

The unique mechanical and electrical properties of graphene also show that it is an interesting material for ultrasonic sensing. The large Young’s modulus of graphene suggests that graphene-based membranes can be easily designed to reach a high resonant frequency, typically in the range of megahertz [11,12,13,14,16,21,24,25]. The superior electrical properties of graphene allow the development of an electrical read-out for electromechanical ultrasonic devices. Ultrasonic detection has been used in medical imaging [26], non-contact sensing [27], non-destructive testing [28], ultrasonic range finding [29] and ultrasound identification [30]. The desired ultrasonic frequency is from 20 kHz and up to GHz dependent on the application. Furthermore, previous work on graphene-based ultrasonic sensors has reported that it can be detected in a vacuum [14]. Apart from ultrasonic sensing, another application of resonators with a resonant frequency less than 200 kHz is to achieve microphones with a good signal-to-noise ratio (SNR) and sensitivity. The resonant frequency in our previous work on a graphene/PMMA capacitive microphone [31] has been observed to be within the audio frequency, which decreases the sensitivity of the microphones. In commercial capacitive microphones, the resonant frequency of diaphragms has been designed to be beyond the audio frequency range. However, in the atmosphere, the vibrational magnitude of the graphene-based ultrasonic sensors can be difficult to be detected due to the presence of air damping.

To date, there has been limited study of graphene-based resonators reported to cover the ultrasonic frequency range between 20 kHz and 1 MHz. To achieve a relatively lower ultrasonic frequency range (below 1MHz), a larger size of graphene-base membranes is required, which increases the complexity of graphene transfer and the difficulty in suspending the graphene-based membrane over the substrate without collapse. To address these two problems, an ultrasonic transducer [20] has been reported to be developed by the transfer of a six-layer graphene membrane onto a supporting frame, which is afterwards manually assembled with the bottom electrode. The air gap formed by the manual assembly of the graphene membrane and substrate is a variable parameter, which might decrease the consistency of device operation. The key in fabricating a graphene-based ultrasonic sensor for a lower ultrasonic frequency is to develop a one-step process to control the air gap in order to avoid the manual assembly process, which can decrease the inconsistency in device fabrication and operation.

In this work, a fully clamped graphene/PMMA closed cavity resonator at a resonant frequency less than 200 kHz will be presented. To avoid the membrane being transferred on the supporting ring and assembled onto the substrate afterwards, a one-step graphene dry transfer process has been developed by our group [15]. A six-layer graphene structure reinforced by 450-nm-thick PMMA has been transferred directly onto the substrate and suspended fully over a closed circular cavity with a diameter of 0.5 mm and formed an air gap of 105 μm. The thin PMMA layer functions not only as the attachment between the graphene and the anchor of the substrate but also as the supporting layer for the graphene to be suspended over the closed cavity. The air gap of 105 μm has been designed to minimize the effect of air damping. The sensor has been actuated mechanically, electro-statically and electro-thermally in the atmosphere. This is the first time that the dynamic resonant characteristics of a graphene/PMMA ultrasonic closed cavity resonator have been determined.

## 2. Materials and Methods

An optical image of the graphene/PMMA ultrasonic closed cavity resonator is shown in Figure 1a. The graphene/PMMA membrane has been transferred onto the silicon dioxide on a silicon substrate with a closed cavity, of which the air gap has been designed to be 105 μm. The squares at the corners of the chip have been patterned and etched into silicon to serve as electrodes. As the cross-section schematic of the device in Figure 1b shows, an air gap of 105 μm has been formed by the suspended membrane and the silicon substrate, which has been measured by a Leica 150x optical microscope. The capacitance between the membrane and the substate has been calculated to be 16.5 fF. The graphene/PMMA membrane and the silicon substrate work as two plates for the capacitive structure. The natural frequency formula for the graphene/PMMA membrane can be determined as follows: (1)$$\begin{array}{cc} t_{\mathrm{eff}} & =t_{g}+t_{p}, \end{array}$$(2)$$\begin{array}{cc} \rho_{\mathrm{eff}} & =\frac{\rho_{g}t_{g}+\rho_{p}t_{p}}{t_{g}+t_{p}}, \end{array}$$(3)$$\begin{array}{cc} A_{m} & =\frac{\rho_{air}R}{3\rho_{\mathrm{eff}}t_{\mathrm{eff}}}, \end{array}$$(4)$$\begin{array}{cc} f_{mn} & =\frac{\beta_{mn}}{2\pi R}\sqrt{\frac{N_{i}+N_{a}}{\rho_{\mathrm{eff}}t_{\mathrm{eff}}\left(1+A_{m}\right)}}, \end{array}$$
where *t* and ρ are the thickness and mass density of the material; $t_{\mathrm{eff}}$ and $\rho_{\mathrm{eff}}$ refer to the effective thickness and effective mass density for the graphene (g)/PMMA (p) bi-layer membrane; *R* is the radius of the membrane; $\rho_{air}$ refers to the air density; $A_{m}$ is the air mass; $N_{i}$ and $N_{a}$ represent the membrane’s built-in tension and actuation tension, which is caused by dynamic actuation, and $\beta_{mn}$ is a dimensionless coefficient of the resonant mode.

The fabrication process of the graphene/PMMA closed cavity ultrasonic sensor is shown in Figure 2. The preparation of the device’s substrate is shown in Figure 2i,ii; the 500 nm silicon dioxide has been deposited onto the silicon substrate. The circular cavity with the diameter of 500 μm, together with three square holes with 100 μm width that serve as electrodes, have been patterned and etched into the silicon dioxide and silicon. The preparation of the graphene/PMMA membrane is shown as follows: (iii) the Kapton tape frame attached to the copper CVD graphene; (iv) the PMMA spin-coated on the CVD graphene; (v) the copper foil etched by ferric chloride; the dry transfer of the graphene/PMMA membrane, including (vi) the graphene/PMMA membrane dry-transferred onto the substrate and the Kapton tape frame peeled off from membrane at the temperature of 140 °C; (vii) the device cooled down in the air. Additionally, the graphene dry transfer method has also been reported in our previous publication [15]. In this work, the success rate of the fabrication process has been 100% over two devices.

## 3. Results and Discussion

### 3.1. Dynamic Actuation

The graphene/PMMA ultrasonic resonator has been actuated mechanically, electro-statically and electro-thermally to characterize its dynamic behavior. For the mechanical actuation, the graphene/PMMA ultrasonic resonator has been placed and attached to the piezoelectric disk. By applying a voltage to the piezoelectric disk, ultrasound vibration has been generated and actuated the substrate of the resonator. For the electro-static actuation, silver paste has been attached to the graphene layer to work as the top electrode. The patterns etched into the silicon with the resistivity of 1–10 Ωcm have been used as the bottom electrodes. The electro-static stress between the graphene membrane and the substrate has been generated by the voltage applied to the top and bottom electrodes. For the electro-thermal actuation, the thermal expansion of the membrane has been actuated by the voltage applied to the silver paste on the graphene layer. The dynamic characteristics have been measured by a Polytec Laser Doppler Vibrometer (LDV). In addition to actuating the resonator through the signal with the frequency sweep, the sine-function signal of the membrane’s resonant frequency has also been applied in order to provide a larger response time for the membrane to be actuated and to improve the accuracy of the displacement of the membrane that has been measured. All the measurements have been conducted on one device at room temperature and in the atmosphere.

#### 3.1.1. Mechanical Actuation

For mechanical actuation, the varying AC voltage from 0.2 V to 3 V and a constant 1 V DC voltage with a frequency sweep from 150 kHz to 220 kHz have been applied to the piezo-disk. The frequency response of the membrane is shown in Figure 3a. The resonant frequency of the membrane has been measured to be around 163.15 kHz ± 0.2% with a side band of around 169.487 kHz. The side band can be explained by the coupling between the membrane and substrate. The frequency peak at around 169 kHz has been observed with the graphene/PMMA membrane stuck on the silicon dioxide substrate anchor under mechanical actuation (Appendix A). The frequency response measured under the frequency sweep at 0.1 V AC and 0.2 V AC seems to be similar, which can be explained by the response time at an ultrasonic frequency of around 163 kHz being too small for the membrane actuated at the lower AC voltages to respond and reach its maximum value. The quality factor at the resonant frequency has been estimated to be 49.45 ± 6.8%.

#### 3.1.2. Electro-Static Actuation

For electro-static actuation, the voltage of constant 1 V DC and varying AC voltage from 4 V and 9 V with a frequency sweep between 120 kHz to 200 kHz have been applied between the membrane and substrate. The frequency response of the graphene/PMMA membrane is shown in Figure 3b. The resonant frequency has been measured to be 158.337 kHz ± 0.4% with the side band observed at 169.265 kHz. The likely explanation for the side band is the coupling between the membrane and substrate. As with the mechanical actuation, the actuation stress (electro-static stress) is vertical to the membrane. In addition, the side band frequency at the electro-static actuation has been observed to be similar to the side band frequency observed from the mechanical actuation (Figure 3a). The quality factor has been observed to be 25.64 ± 5.8% at the resonant frequency.

#### 3.1.3. Electro-Thermal Actuation

For electro-thermal actuation, the frequency response of the resonator actuated by increasing 1 V to 9 V AC and 1 V DC voltage applied to the silver paste on the graphene/PMMA membrane with the frequency range from 140 kHz to 220 kHz is illustrated in Figure 3c. The resonant frequency has been observed to be around 158.965 kHz ± 1.9% with the side band of around 187.851 kHz. The side band can be explained by the transition between the (1,1) mode and (0,2) mode (Appendix A). Under the electro-thermal actuation, the membrane has been heated when the AC voltage is applied and the transition between the (1,1) mode and (0,2) mode can result from thermal stress in the membrane. Such a transition has not been observed in mechanical and electro-static actuation. Unlike the other two actuation methods, where the actuation stress has been out-of-plane, in the case of electro-thermal actuation, the thermal expansion generated by the Joule heating has been in-plane. The likely explanation is that the in-plane actuation stress through the membrane has not generated the coupling between the membrane and substrate. In the case of the other two actuation methods, the coupling between the membrane and substrate dominates the vibration at the side band frequency and a transition with a smaller amplitude has not been observed. At the resonant frequency, the quality factor has been detected to be 34.42 ± 15.8%.

### 3.2. Sensitivity of Vibration Amplitude

The vibration amplitude of the graphene/PMMA membrane over the closed cavity is shown in Figure 4. The membrane has been actuated by a sinusoidal signal at the resonant frequencies corresponding to different actuation methods. The amplitude of the membrane has been observed to be linear with the increasing AC voltage under the mechanical and electro-static actuation, as illustrated in Figure 4a,b. In the case of the electro-thermal actuation, the graphene/PMMA membrane has been actuated by the thermal stress that has been generated by Joule heating. The thermal stress is linear with the Joule heating and thus is quadratic with the input AC voltage. As shown in Figure 4c, the quadratic relation between the amplitude and input AC voltage from 1 V to 8 V has been observed. At a voltage of 9 V AC, the amplitude, which has not been shown to fit with the parabola function, can be explained by the membrane’s resonant frequency being shifted by the increasing AC voltage. At 9 V AC, the resonant frequency of the membrane over the closed cavity has been measured to be 161.914 kHz, with the frequency shift of around 5 kHz away from the actuated sinusoidal signal at a frequency of 156.914 kHz (Figure 4). The amplitude at the frequency with around 5 kHz shifted from the resonant frequency is smaller than the amplitude measured at the resonant frequency.

The dynamic behavior of the graphene/PMMA closed cavity ultrasonic sensor is summarized in Table 1. Under electro-static actuation, the explanation for the small amplitude measured in the frequency sweep is the air gap of around 105 μm, which forms a small capacitance between the membrane and substrate. The measured resonant frequency has been observed to change with the actuation method. In the cases of electro-static and electro-thermal actuation, the measured resonant frequency is smaller compared to the mechanical frequency, which can be explained by capacitive softening [32,33,34] and electro-thermal softening [35].

### 3.3. Frequency Shift and Quality Factor

In mechanical and electro-static actuation, the change in frequency shift and quality factor versus the input signal has been detected to be relatively small compared to the electro-thermal case, as shown in Figure 5a–c.

In the case of the electro-thermal actuation, the change in quality factor can be temperature-related. The frequency shift at resonance is evident in the frequency response (Figure 3c). The relationship between the frequency shift and the AC voltage is plotted in Figure 5c. The resonant frequency at 9 V AC has been upshifted to be 3.8% from the frequency at 2 V AC. The upshift of the resonant frequency as the AC voltage increases can be a result of the negative thermal expansion coefficient of graphene [36]. Graphene shrinks as its temperature rises and, therefore, the resonant frequency increases with the rising AC voltage [16]. The fitting (red dash) of the frequency shift corresponds to $V_{ac}^{\frac{2}{3}}$. The nonlinearity of the frequency shift can be explained by the air damping inside the perfectly sealed closed cavity.

As shown in Table 1, the quality factor when the membrane is actuated mechanically has been observed to be the maximum among the three actuation methods. The piezo-electric disk has been directly in contact with the substrate during mechanical actuation and therefore the input ultrasonic energy has been the largest among the three actuation methods. The quality factor measured under electro-static actuation has been measured to be the minimum among the three actuation methods, which is related to the smallest displacement observed compared to the other two actuation methods. The air gap of 105 μm results in a capacitance of 16.5 fF and the signal generated by the electro-static stress between the membrane and substrate is relatively small compared to the other two actuation methods.

The change in the quality factor has been studied in the resonator under the electro-thermal actuation. The quality factor has been measured to increase from around 36 to 40 when the AC voltage rises from 2 V and 3 V. A decrease in the quality factor has been observed when the AC voltage changes from 3 V to 8 V. A small increase in the quality factor was measured when the AC voltage increased from 8 V to 9 V, as shown in Figure 5c. Unlike the mechanical and electro-static actuation, a frequency upshift has been observed in the resonator under electro-thermal actuation. The decrease in the quality factor suggests that the energy dissipated in the resonator is larger than the energy stored at the resonant frequency [37], which can be explained by the higher damping [16] or greater surface loss [38] of the energy as a higher AC voltage is applied to the membrane. The boost in the thermal gradient in the membrane with an increasing AC voltage might enhance the thermoelastic damping, which increases the dissipation [37]. Additionally, the possible surface stress increase with the rising temperature might enlarge the surface loss, which results in energy dissipation [38,39].

### 3.4. Mode Shape

The mode shapes at the resonant frequencies obtained by different actuation methods are shown in Figure 6. The observation of (1,1) at the resonant frequencies by the three actuation methods has been caused by the closed cavity design and the impermeability of graphene [40]. The air leakage is extremely small as the graphene/PMMA membrane has sealed the closed cavity perfectly. Thus, the (0,1) mode, which requires a large change in the air volume inside the cavity, has been prevented and not been observed. Figure 6a–c are placed at the same x–y plane to compare the orientations under different actuation schemes. The orientation of the (1,1) mode shape has been observed to be similar in the mechanical and electro-static actuation, which can be explained by the direction of the mechanical stress and electro-static stress, which is vertical. In the case of electro-thermal actuation, the orientation of the (1,1) mode shape has been related to the position of the membrane electrodes.

### 3.5. Strain Analysis

The overall tension and strain can be derived from Equation (Equation 4) and the results are shown in Table 2. In the case of the mechanical actuation, the tension has been estimated to be the largest among the different actuation methods.

## 4. Conclusions

This is the first time that a graphene-based closed cavity ultrasonic resonator has been fabricated and actuated in the atmosphere successfully. Using the graphene dry transfer method with Kapton tape as the supporting frame, developed by our group, a graphene/PMMA closed cavity sensor at a resonant frequency of around 160 kHz has been fabricated. The graphene/PMMA closed cavity resonator has been actuated mechanically, electro-statically and electro-thermally. The amplitude of the membrane has been observed to be linear with AC voltage for the mechanical and electro-static actuation and quadratic with AC voltage for the electro-thermal actuation. The membrane has been observed to exhibit a (1,1) mode at the resonant frequencies. The membrane can be tuned by up to 4% by varying the AC voltage via the electrodes connected to the graphene/PMMA membrane, and a nonlinear frequency shift under electro-thermal actuation has been detected. The strain in the membrane under the three actuation methods has been estimated to be around 0.08%. The device shows the possibility of applying graphene as an ultrasonic detector and opens a door to the fabrication of graphene-based ultrasonic sensors at a lower ultrasonic frequency of less than 200 kHz.

## Figures and Tables

**Figure 1 micromachines-14-00810-f001:**
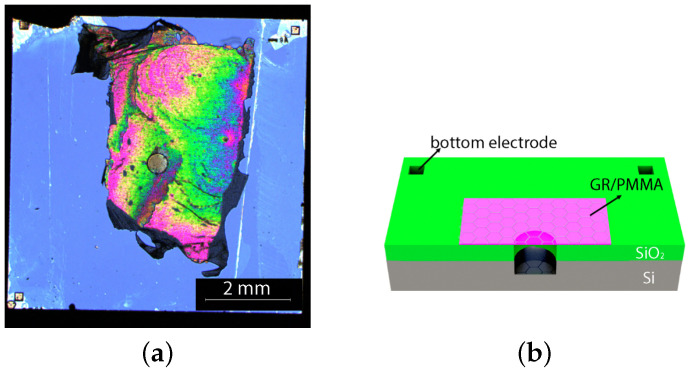
The optical image (**a**) and cross-section schematic (**b**) of the closed cavity resonator with 105 µm gap.

**Figure 2 micromachines-14-00810-f002:**
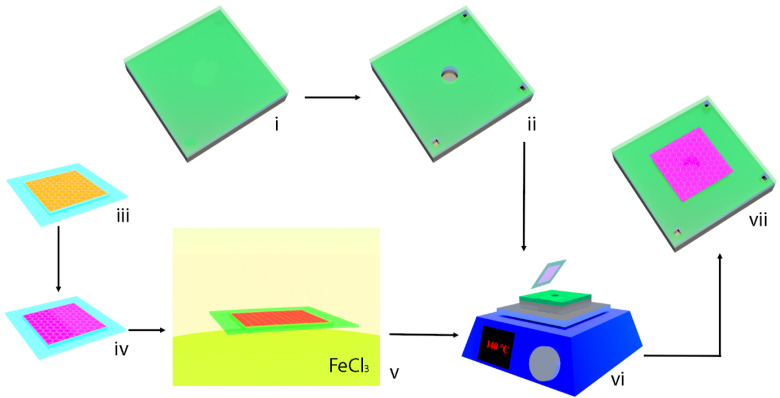
The fabrication schematic of the graphene/PMMA closed cavity ultrasonic sensor.

**Figure 3 micromachines-14-00810-f003:**
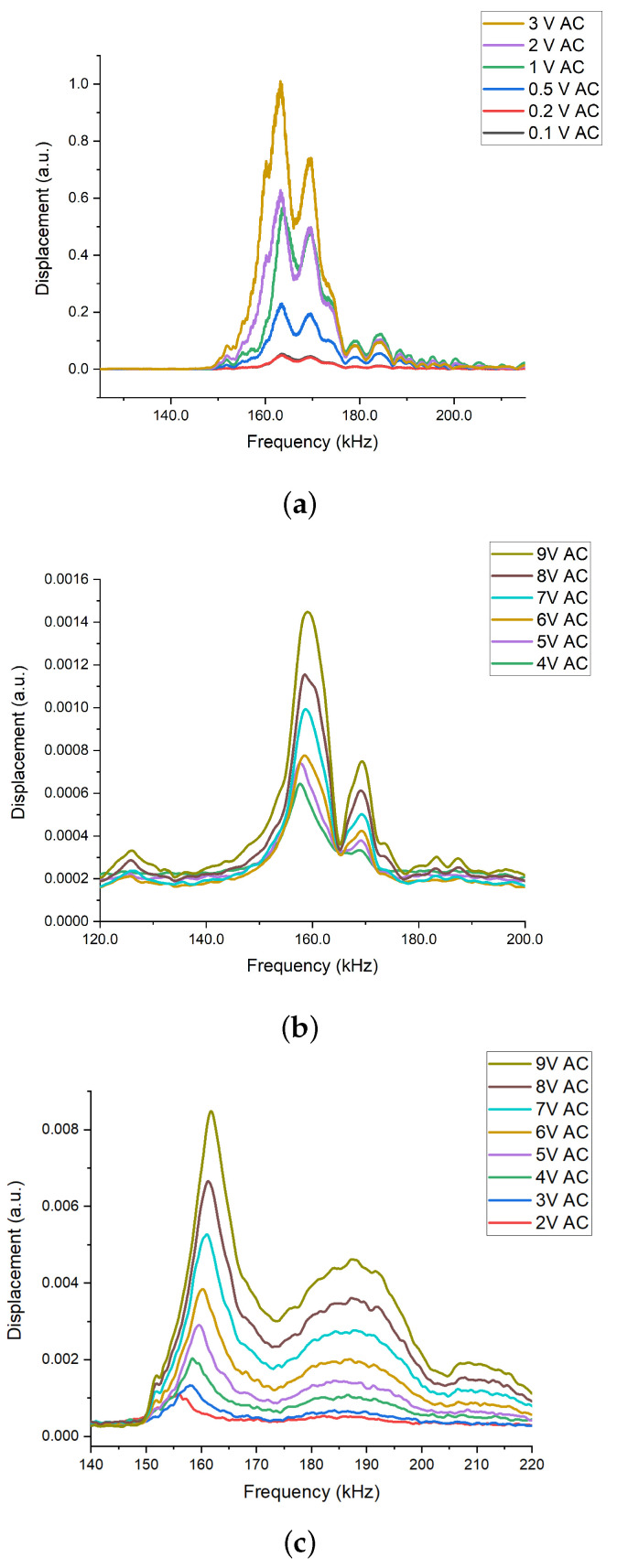
The frequency response of the membrane under (**a**) mechanical actuation with the input voltage from 0.1 V to 3 V AC and 1 V DC, as well as by the frequency sweep from 150 kHz to 220 kHz; (**b**) electro-static actuation with the voltage of constant 1 V DC voltage and varying AC from 4 V to 9 V with the frequency sweep between 120 kHz and 200 kHz; (**c**) electro-thermal frequency sweep signal with 2–9 V AC and 1 V DC between 140 kHz and 220 kHz.

**Figure 4 micromachines-14-00810-f004:**
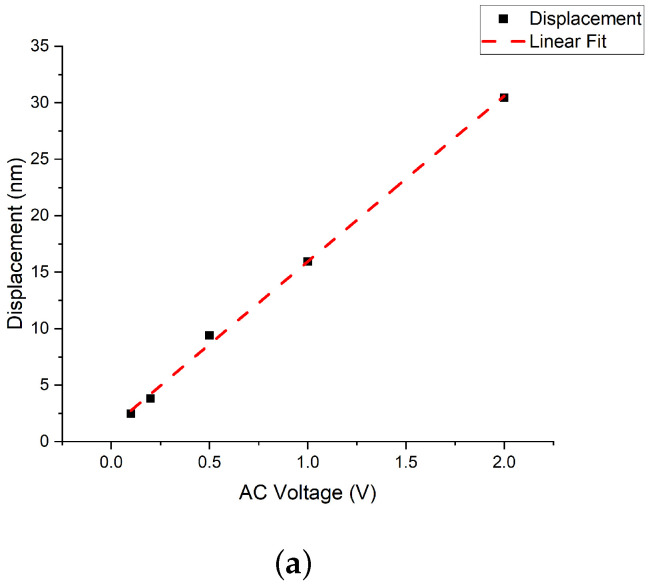
The amplitude of the membrane under (**a**) mechanical actuation (0.1 V AC to 2 V AC and constant 1 V DC) at 163.156 kHz with linear fitting; (**b**) electro-static actuation at 158.640 kHz with signal of the AC voltage changing from 1 V to 9 V and constant 1 V DC, with linear fitting; (**c**) electro-thermal actuation at 156.914 kHz with the voltage of 1–9 V AC and 1 V DC along with parabola fitting.

**Figure 5 micromachines-14-00810-f005:**
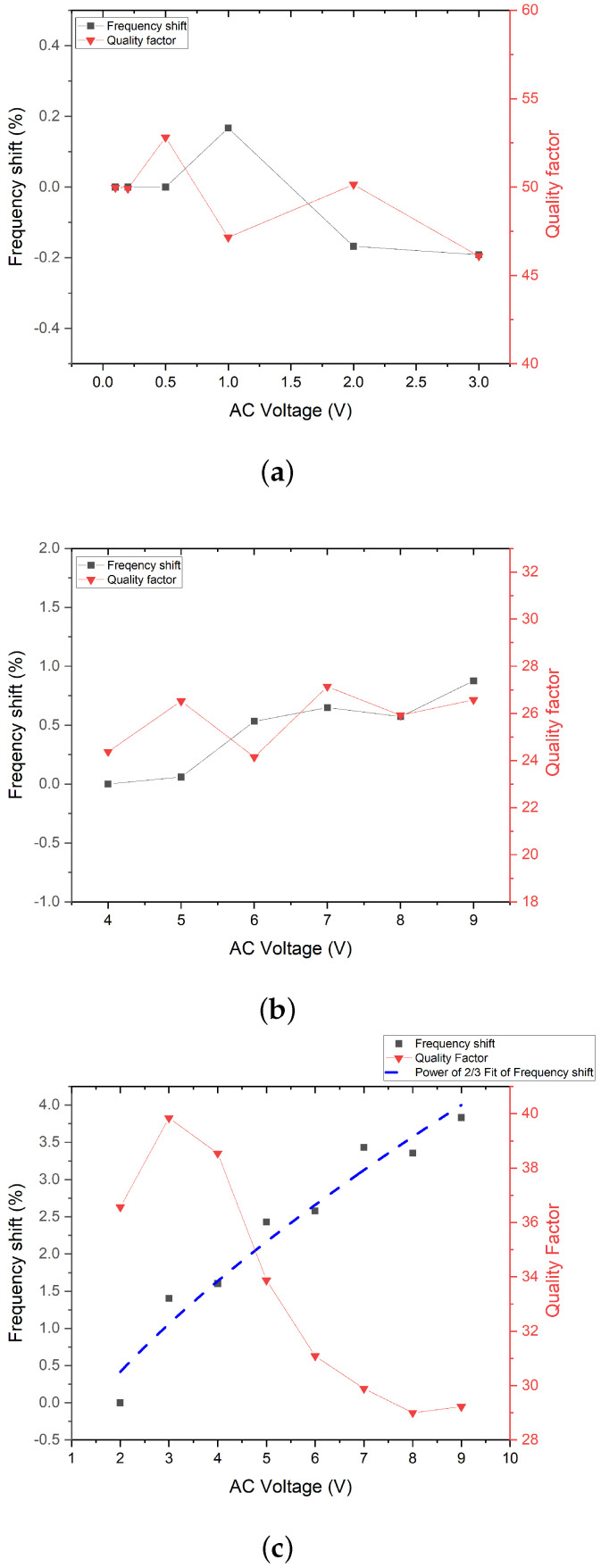
The frequency shift and quality factor of graphene/PMMA resonator under (**a**) mechanical actuation (**b**) electro-static actuation; (**c**) electro-thermal actuation.

**Figure 6 micromachines-14-00810-f006:**
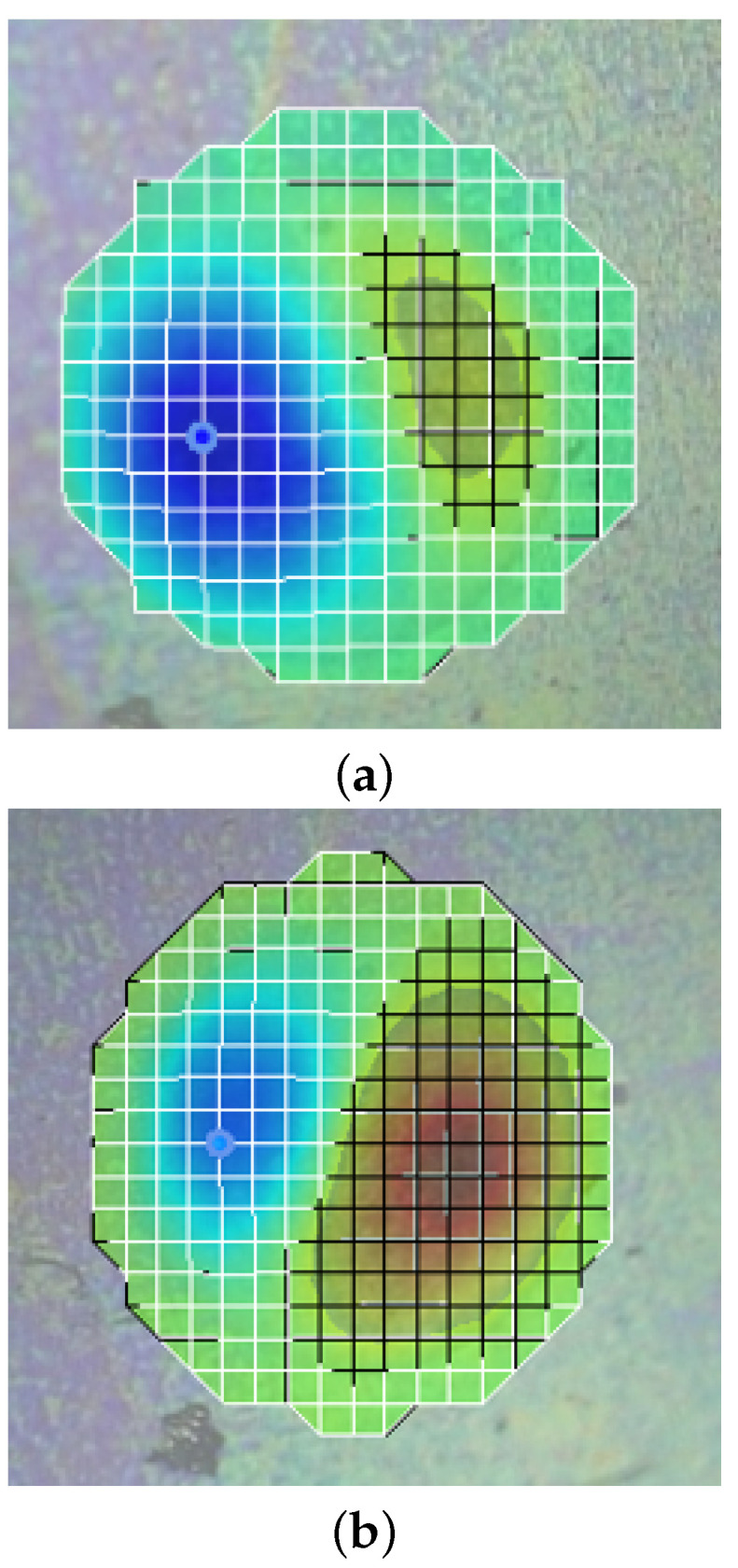
The mode shape of graphene/PMMA membrane over closed cavity resonator at resonant frequencies under (**a**) mechanical actuation (0.5 V AC, 1 V DC); ((**b**) electro-static actuation (3 V AC, 1 V DC); (**c**) electro-thermal actuation (3 V AC, 1 V DC).

**Table 1 micromachines-14-00810-t001:** The dynamic characteristics of graphene/PMMA closed cavity ultrasonic sensor.

Actuation Methods	Measured Resonant Frequency	Quality Factor	Actuated Sinusoidal Signal Frequency	Varying Input Signal Range of Sinusoidal Signal	Sensitivity of Vibration Amplitude Actuated by Sinusoidal Signal
Mechanical	163.150 kHz ± 0.2%	49.45 ± 6.8%	163.156 kHz	0.1 V to 2 V AC	14 nm/V
Electro-static	158.337 kHz ± 0.4%	25.64 ± 5.8%	158.640 kHz	1 V to 9 V AV	0.01 nm/V
Electro-thermal	158.965 kHz ± 1.9%	34.42 ± 15.8%	156.914 kHz	1 V to 9 V AV	0.002 nm/V${}^{2}$

**Table 2 micromachines-14-00810-t002:** Overall tension and strain in the graphene/PMMA membrane deducted from the measured resonant frequency.

Actuation Methods	Frequency (kHz)	Tension (N/m)	Strain (%)
Mechanical	163.150	3.00	0.0813
Electro-static	158.384	2.83	0.0766
Electro-thermal	158.965	2.85	0.0772

## Data Availability

The data used to support the study are available upon request to thecorresponding authors.

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
