# Peer review of "A Closed Cavity Ultrasonic Resonator Formed by Graphene/PMMA Membrane for Acoustic Application"

_micromachines, 2023, doi:10.3390/mi14040810_

Round 1

Reviewer 1 Report

A graphene/PMPA closed cavity resonator with resonant frequency at around 160 kHZ has been fabricated in this paper. Some tests have been done for the resonator.
The writing of the article is concise and clear, however,  the solution of the following problems will improve the article.

1. Please describe the advantages of this device, and the limitations of this study?

2.Please elaborate the processes of fabrication of graphene/PMPA closed cavity closed cavity ultrasonic sensor.

3.The references are too old, please cite some references within 5 years.

4. How many experiments are the scatterplots from? Why are there no upper and lower deviations on the graph?

5. Why some words are hyphenated in the paper? For example: ‘tempera-ture’,  ‘sen-sor’.

Author Response

Dear Reviewer,

Thanks for your constructive comments. Please see the response in the attachment. 

Best wishes, Jing

Reviewer 2 Report

For section 3.2, an additional figure to show the input sinusoidal signals and measured membrane displacement synchronized in time could be added. Corresponding discussion in magnitude and possible phase shift could also be added.

Some of the written English could be improved via proof reading

Presentation of figures could be improved to be clearer

Author Response

(The authors gave the same response as above.)

Round 2

Reviewer 1 Report

The authors have completed the modification and  I recommend this paper should be accepted.